# Maternal High-Energy Diet during Pregnancy and Lactation Impairs Neurogenesis and Alters the Behavior of Adult Offspring in a Phenotype-Dependent Manner

**DOI:** 10.3390/ijms23105564

**Published:** 2022-05-16

**Authors:** Kamila Fabianová, Janka Babeľová, Dušan Fabian, Alexandra Popovičová, Marcela Martončíková, Adam Raček, Enikő Račeková

**Affiliations:** 1Institute of Neurobiology, Biomedical Research Center, Slovak Academy of Sciences, Šoltésovej 4, 040 01 Košice, Slovakia; popovicova@saske.sk (A.P.); martoncikova@saske.sk (M.M.); racek@saske.sk (A.R.); racekova@saske.sk (E.R.); 2Centre of Biosciences, Institute of Animal Physiology, Slovak Academy of Sciences, Šoltésovej 4-6, 040 01 Košice, Slovakia; kubandova@saske.sk (J.B.); fabian@saske.sk (D.F.)

**Keywords:** maternal nutrition, obesity, subventricular zone, rostral migratory stream, neurogenesis, high-energy diet, cell proliferation, cell degeneration, nitric oxide

## Abstract

Obesity is one of the biggest and most costly health challenges the modern world encounters. Substantial evidence suggests that the risk of metabolic syndrome or obesity formation may be affected at a very early stage of development, in particular through fetal and/or neonatal overfeeding. Outcomes from epidemiological studies indicate that maternal nutrition during pregnancy and lactation has a profound impact on adult neurogenesis in the offspring. In the present study, an intergenerational dietary model employing overfeeding of experimental mice during prenatal and early postnatal development was applied to acquire mice with various body conditions. We investigated the impact of the maternal high-energy diet during pregnancy and lactation on adult neurogenesis in the olfactory neurogenic region involving the subventricular zone (SVZ) and the rostral migratory stream (RMS) and some behavioral tasks including memory, anxiety and nociception. Our findings show that a maternal high-energy diet administered during pregnancy and lactation modifies proliferation and differentiation, and induced degeneration of cells in the SVZ/RMS of offspring, but only in mice where extreme phenotype, such as significant overweight/adiposity or obesity is manifested. Thereafter, a maternal high-energy diet enhances anxiety-related behavior in offspring regardless of its body condition and impairs learning and memory in offspring with an extreme phenotype.

## 1. Introduction

Obesity is one of the biggest and most costly health challenges the modern world encounters [1,2,3]. The incidence of obesity has increased significantly in both developed and developing countries over the last thirty years and is already a major public health challenge in several countries [4,5,6,7,8,9]. What is worse, obesity affects not only adults but also children and adolescents [10]. In addition to the well-described effects of being extremely overweight or obese on the cardiovascular system and metabolic processes [2,3], new epidemiological and experimental studies are confirming the adverse effects of obesity and obesity-related metabolic disorders on the central nervous system (CNS) [11,12,13,14,15,16,17].

In general, obesity can be defined as the physiological state in which body fat has accumulated to a volume that is damaging to overall health [18] and in particular it is a result of overfeeding/overconsumption of food rich in energy, especially a high-caloric diet. High-caloric diet (HCD) is a general concept used for naming of several dietary schemes, which contain excessive volumes of calories derived from enriched proteins, carbohydrates, fat or some combination of these macronutrients. Recent studies in rats and mice have shown that these kinds of nutritional regimes can lead to increased somatic growth and delayed reflex ontogeny during lactation [19], cognitive impairment [20,21,22,23,24], enhanced risk of Alzheimer’s disease [25,26,27,28], anxiety and depression [24,29,30], alterations in learning and memory tasks [31,32,33] or impairment of brain insulin signaling linked with neuroinflammation [34,35]. In humans, HCDs are mainly related to metabolic and cardiovascular diseases, but there is growing evidence that diets rich in fat and carbohydrates at the same time in humans are accompanied by elevated risk of Alzheimer’s disease [36,37], Parkinson’s disease [38], anxiety and depression [39,40,41], dementia [36,42,43] and various mood disorders [39,44].

Most of all HCDs, the high energy diet (HED) based on excessive intake of simple carbohydrates, has been proven to be one of the outstanding weight gain agents [45]. Furthermore, a nutrition rich in simple carbohydrates has been connected to cognitive disturbances, hippocampal malfunction [46,47], increased Alzheimer’s disease-associated pathologies [48] and mood disorders involving anxiety and depression [49].

Profound evidence suggests that the risk of evoking metabolic syndrome or obesity may be affected very early in development, in particular through fetal and/or neonatal overfeeding [50,51,52,53,54]. Past rodent studies suggest that maternal exposure to HCD during pregnancy and breastfeeding is a prominent source of modifications of normal infant programming, both in animals and humans, and can impair brain development [55,56,57]. HCD fed dams often struggle with advanced metabolic disorders, causing high level of lipids and other nutritious components in blood. These macronutrients can be delivered to developing offspring via the placenta or breast milk [58,59,60,61]. Consequently, HCD in dams can affect the development and neural plasticity of the brains of infants and these neuronal alterations can persist after birth in the absence of unhealthy nutrition and have long-term consequences [19,62,63,64]. Moreover, evidence from epidemiological studies indicates that maternal overfeeding during pregnancy or lactation periods has a profound impact on adult neurogenesis in the offspring [56,63,65,66,67].

Adult neurogenesis is a complex process, involving the proliferation of neural progenitor cells and their consecutive migration, differentiation and functional integration into the pre-existing circuitry [68]. Neural progenitor cells possessing the potential to give rise to neurons in vitro are presumably scattered all over the adult mammalian CNS. Nevertheless, there are only two regions in the adult brain, in which newborn neurons are regularly observed: the subgranular zone (SGZ) in the dentate gyrus of the hippocampus [69] and the subventricular zone (SVZ) of the lateral ventricles [70]. Lately, the hypothalamus was confirmed as another area, in which new glial cells and neurons are gradually arising from proliferating neural progenitor cells, although at a much lower rate than in the SVZ and SGZ [71]. The SVZ, covering the lateral ventricle, is the largest neurogenic zone in the adult mammalian brain, comprising various cell populations including astrocyte-like cells and neuroblasts. The adult mouse SVZ is approximately four to five cell diameters thick. Neuronal precursors born within the SVZ migrate through a tangential network in the lateral wall of the lateral ventricle [72] and then turn towards the rostral migratory stream (RMS), which enters the olfactory bulb (OB). The mouse RMS is a thick U-shaped column, approximately 5 mm in length, which arches the anterior horn of the lateral ventricle and the OB. The RMS can be divided into three topographically consecutive anatomical parts: the vertical arm, the elbow and the horizontal arm [73]. Both the SVZ and RMS of rodents undergo gradual and significant thickness reduction from birth to adulthood [74].

Adult neurogenesis can be influenced by many intrinsic and extrinsic factors, including nutrition [75,76,77]. The periods of prenatal and early postnatal life appear to be the critical gap when developmental programming via maternal nutrition can impact neurogenesis and neurobehavioral development of the offspring [55,56,62,78]. Recently, several studies have reported that the offspring of dams which have been fed long-term with HCD have altered hippocampal development with decreased neurogenesis, decreased apoptosis, and decreased neuronal differentiation [20,22,56,67,79], and also neurobehavioral changes [57,78] which can be related to changes in the serotonergic and gabaergic neurotransmitter systems [63]. Perinatal nutrition also modulates hypothalamic neurogenesis [65,80,81]. 

Despite the effect of the maternal HCD during pregnancy or lactation periods on adult hippocampal neurogenesis being well established, little is known about the impact of perinatal dietary regimes of dams on adult neurogenesis in the SVZ/RMS olfactory neurogenic region. Thus, the aim of the present study was to investigate the impact of the maternal high-energy diet during pregnancy and lactation periods on adult neurogenesis in the SVZ/RMS neurogenic region of the offspring and on some behavioral tasks including memory, anxiety and nociception. To assess the effects of maternal HED on neurogenesis in the SVZ/RMS, proliferative activity was observed using the proliferation marker Ki-67 and cell death was monitored using the degeneration cell marker Fluoro-Jade C (FJC). Nitric oxide (NO) producing neurons located directly in the SVZ/RMS were visualized by NADPH-d (NADPH-diaphorase) histochemical staining. The shape and the thickness of the SVZ/RMS were evaluated on hematoxylin-eosin stained sections. Behavioral tests were performed to analyze (1) locomotor activity and anxiety behavior in the open field test, (2) learning and memory processes in the Barnes maze and (3) nociceptive sensitivity in the hot plate test.

## 2. Results

An intergenerational dietary model employing overfeeding of experimental mice during prenatal and early postnatal development was applied to acquire mice with various body conditions. The result of this diet was the origin of offspring (filial generation, F1) with an elevated predisposition for the development of obesity in adulthood. The F1 mice were divided into four groups:➢CN—control group of mice characterized by normal weight (±20 g) and physiological percentage of body fat deposits (7–8%).➢CL—control group of mice characterized by reduced weight and reduced body fat deposits (<7%).➢EXN—experimental group of mice characterized by normal weight and moderately elevated body fat deposits (8–11%), obesity-induction resistant experimental mice.➢EXF—experimental group of mice characterized by markedly elevated weight (±24 g) and body fat deposits (>11%).

### 2.1. The SVZ/RMS Thickness Is Increased in Obese Mice

Although the common morphological appearance of the mouse SVZ/RMS was not impacted in mice with various amounts of body fat, our data showed significant changes in the SVZ and RMS thickness in mice with various body conditions in comparison with controls (Table 1, Figure 1). In EXF mice, we have revealed a significant increase in the thickness of the SVZ and of the RMS caudal parts—the vertical arm and the elbow (Figure 1E). In EXN mice and in lean controls, changes in the thickness of neurogenic areas concerned the RMS vertical arm and elbow, the thickness of which was significantly increased when compared to controls (Figure 1C,D).

### 2.2. Adult Neurogenesis in the SVZ/RMS Is Altered in Mice with Elevated/Decreased Body Fat and Weight in a Phenotype-Dependent Manner

#### 2.2.1. Cell Proliferation

Microscopic evaluation showed differences in the density of Ki-67+ cells within the SVZ/RMS neurogenic region in mice with various body conditions (Figure 2) Ki-67 labeled cells were highlighted by brown stained nuclei. Even at this level of observation, it was obvious that the number of proliferating cells inside the migratory stream is markedly reduced in EXF mice as well as in CL mice in comparison with control mice. The reduction in proliferating cells density in EXF and CL mice was even more striking within the SVZ (Figure 2).

Microscopic observations were confirmed with subsequent quantitative analysis, which pointed out that the number of Ki-67+ cells was significantly lower in the SVZ as well as in all three anatomical parts of the RMS in the EXF mice in comparison with controls. Significant decline in the proliferating cell number was documented also in the SVZ and in the RMS elbow of CL mice when compared to control mice (Table 2). Besides that, quantitative analysis showed a significant decrease in the total number of Ki-67+ cells within the whole SVZ/RMS neurogenic region in mice with elevated/decreased body fat and weight in comparison with control mice (Figure 3).

Moreover, in the CL and EXF mice we have observed a different pattern of dividing cell distribution along the SVZ/RMS area. In intact rodents, the density of Ki-67+ cells reduces in the caudo–rostral direction, with a maximum close to the lateral ventricle and a minimum inside the OB. In EXN mice, this proliferation scheme was sustained. On the contrary, in CL and EXF mice, the proliferating cell number was higher in the RMS elbow in comparison with the SVZ.

#### 2.2.2. Cell Degeneration

In order to map the distribution of degenerating cells within the SVZ/RMS neurogenic area, we have employed FJC histochemistry. FJC+ cells were present in the SVZ/RMS area of all groups examined. Qualitative, fluorescence microscopic analysis showed higher density of FJC+ cells in both SVZ and RMS of mice with elevated/lowered body fat and weight in comparison with control mice (Figure 2). Consecutive quantitative analysis of degenerating cell number in the SVZ/RMS was in accordance with our morphological findings (Table 3). In lean controls (CL), the number of FJC+ cells were significantly higher in the SVZ as well as in all three anatomical parts of the RMS examined, when compared to normal controls. In the EXF mice we observed a significantly higher number of FJC+ cells within the SVZ in comparison with the control animals. Interestingly, in the RMS vertical arm of EXF mice a slightly lower number of FJC+ cells were documented when compared to controls, whilst in the RMS elbow and horizontal arm, the degenerating cell number in the EXF mice was on the same level as in the controls (Table 3).

The total number of FJC+ cells within the entire SVZ/RMS neurogenic area was significantly higher in EXF mice as well as in CL mice when compared to controls. Surprisingly, in comparison with control mice, the increase in the total number of FJC+ cells was more prominent in CL mice than in EXF mice (Figure 4).

#### 2.2.3. Cell Differentiation

To map the distribution of nitrergic (NO producing) cells inside the SVZ/RMS neurogenic area in mice with different body conditions, NADPH-d histochemistry was used. NADPH-d+ cells were present in all evaluated anatomical parts in all groups of mice examined. The labeled cells occurred mainly in the SVZ in both control and experimental animals and a few NADPH-d+ cells were also detected in individual anatomical parts of the RMS. The appearance of labeled cells was similar in all groups of mice examined. NO-producing cells were darkly labeled and typified by their robust oblong or elliptical body. Multiple massive and rough processes of the labeled bipolar cells could also be observed (Figure 2).

Quantitative analysis showed that the number of NADPH-d-positive cells in EXF mice was higher in the SVZ as well as in the RMS individual parts and this increase was significant in the SVZ, in the elbow and in the horizontal arm of the RMS in comparison with controls (Table 4).

Regarding the total number of NADPH-d+ cells within the entire SVZ/RMS neurogenic region, quantitative analysis showed a different trend of alterations in the number of nitrergic cells in EXF and in CL mice when compared to control mice (Figure 4). In the EXF mice, the total number of NADPH-d+ cells were significantly higher in comparison with control animals. Contrary, in the CL mice, the total number of NADPH-d+ cells were significantly lower when compared to controls (Figure 5).

### 2.3. Locomotor Activity Is Reduced in Mice with Elevated/Lowered Body Fat and weight

The open field test and Barnes maze test showed significant reduction in locomotor activity in mice with elevated/lowered body fat and weight (Table 5 and Table 6). The total distance travelled in the open field (OF) as well as in the Barnes maze was significantly shorter in EXF mice and this distance was even lower in CL mice when compared with controls. A decline in the total distance that mice travelled was observed in the EXN mice, although this decrease was not significant. Statistical analysis proved a slight, not-significant decrease in the mean velocity of mice from all three groups examined (CL, EXN, and EXF) in comparison with control mice.

### 2.4. Overfeeding during Prenatal and Early Postnatal Development Enhances Anxiety-Related Behavior

There were no significant differences found in the number of grooms and in the time spent grooming among the control and experimental groups examined (CL, EXN and EXF), but the latency to first grooming was significantly longer in the CL and EXF mice. The CL mice spent significantly shorter periods of time rearing when compared to the CN group. Two-way ANOVA showed that mice with elevated/decreased body fat and weight from all three groups examined (CL, EXN and EXF) reared significantly less often than the controls and this decline was most apparent in CL mice. In addition, the CL mice and EXF mice spent significantly longer periods of time freezing than control mice. In EXN and EXF mice, the time spent near to the walls of the OF was significantly enhanced and thigmotaxis (ratio central vs periphery) was significantly lowered relative to controls. The mice from all three groups examined (CL, EXN and EXF) crossed the OF middle point significantly fewer times in comparison with control mice and this decrease was most prominent in CL mice. Neophobia as measured by the latency to leave the peripheral compartment for the first time was significantly enhanced in CL and EXF mice. Our results also showed a significant increase in defecation in CL and EXF mice. Behavioral scores for anxiety parameters in CN, CL, EXN and EXF group are shown in Table 7.

### 2.5. Mice with Various Amounts of Body Fat Show Comparable Performances in the Hot Plate Test

The hotplate device was applied to investigate the thermal pain responses in mice with various body conditions. Although mice from all experimental groups (CL, EXN and EXF) showed a little shorter hot-plate withdrawal latency than the control group, no significant differences in latencies to lick a paw or jump off were observed (Figure 6).

### 2.6. Maternal HED Exposure Impairs Spatial Memory and Learning in a Phenotype-Dependent Manner

Based on the Barnes maze test outcomes, there were significant intergroup differences in mice with various body conditions in spatial learning and memory performance (Figure 7). Compared with controls, mice with elevated/lowered body fat and weight (EXF and CL mice) entered significantly more incorrect holes, which matches an increased error ratio in the Barnes maze. Moreover, both EXF and CL mice showed significantly increased latency to locate the escape box during testing. These results suggest that mice with significantly elevated/decreased body fat and weight have impaired learning and memory when compared to control mice. On the other hand, EXN mice showed Barnes maze performances, which were comparable to controls.

## 3. Discussion

The perinatal period, that includes pregnancy and lactation, is crucial for brain development in mammalian species. Recently, several studies have reported that delicate processes of brain formation can be rewired by environmental cues occurring at this sensitive stage of development. Maternal overnutrition during the perinatal period is known to cause multiple disturbances in brain development; of which altered neurogenesis is one manifestation. Additionally, a poor quality of early nutritional environment (i.e., excessive intake of fat, sugar or energy) can alter the behavior of offspring and increase susceptibility to diseases and neurocognitive disorders. 

It is well established that maternal overnutrition during pregnancy and lactation leads to changes in adult hippocampal and hypothalamic neurogenesis in offspring, and these changes could result in cognitive impairments or emotional disorders such as anxiety and depression [46,47,63,65,66,80]. Up to now, the effect of maternal overnutrition, on SVZ/RMS morphology and SVZ neurogenesis in adult offspring has not been explored. The objective of the present study was to determine whether similar HED-induced changes in neurogenesis can be demonstrated in the SVZ/RMS neurogenic region and also whether these changes are accompanied by alterations in behavioral performance in tasks related to cognition and anxiety. We employed an intergenerational dietary model, which imitates the current human lifestyle, characterized by a high-calorie and high-sugar diet, to investigate its possible impact on adult neurogenesis in the SVZ/RMS neurogenic region and the behavior of adult mice.

### 3.1. The SVZ/RMS Thickness Is Increased in Obese Mice

In mice, the thickness of the SVZ and RMS decreases gradually but significantly from birth to adulthood [82]. Besides that, the overall appearance and size of rodent SVZ and RMS can be affected by various extrinsic factors [74]. Since the impact of aging [83] and particular experimental interventions [74,84] on the morphology of the SVZ/RMS neurogenic niche is well established, the effect of dietary patterns or obesity has not been studied yet.

Our findings showed that the general morphological appearance of the SVZ/RMS is not impacted in mice with various amounts of body fat. On the other hand, we have observed an increase in the thickness of both SVZ and RMS individual parts in all experimental groups. In general, the increase in the SVZ/RMS thickness can result from an imbalance in SVZ/RMS proliferation, migration and cell death caused by adverse perinatal influences. Dramatic increase in the size of the SVZ/RMS due to accumulation of neuronal precursors was documented following several interventions [84,85,86]. In this study, changes in the SVZ/RMS thickness did not reflect changes in the number of proliferating and degenerating cells, so we assume that there is another reason for the increased SVZ/RMS size. 

It is very likely that the increase in the SVZ/RMS thickness in mouse with elevated/decreased body fat can be evidence of retardation in the SVZ/RMS development. Overnutrition can be related to adverse perinatal events that affect structural brain development [87,88,89,90] or cause delayed maturation of the SVZ/RMS [74].

Besides numerous studies documenting the impact of perinatal nutrition on morphology of the hippocampus [91,92,93,94], studies dealing with the effect of maternal nutrition on the morphology of the SVZ/RMS neurogenic niche are very sparse. Bertrand et al. [95] have found that fatty acid deficiency during gestation increased the thickness of the ventricular zone in 19 day old embryos, but at the same time, no significant difference in the thickness of the SVZ was observed. According to the authors, reported morphological changes can be a manifestation of deceleration or retardation of neurogenesis in the (n-3) fatty acid embryos starting at earlier stages of development.

### 3.2. Adult neurogenesis in the SVZ/RMS Is Altered in Mice with Elevated/Decreased Body Fat and Weight in a Phenotype-Dependent Manner

The primary goal of this study was to determine whether maternal HED during pregnancy and lactation periods affects individual processes of SVZ/RMS neurogenesis, specifically cell proliferation, cell degeneration and cell differentiation. As previously reported, a high-fat diet (HFD) and HED during gestation and lactation modifies hippocampal neurogenesis including an elevation in the number of dying/degenerating cells, a decline in the total number of doublecortin expressing cells and reduction in the numbers of newly born cells in the offspring’s dentate gyrus [67,79,96,97,98]. In recent years, besides the hippocampal neurogenesis, the impact of maternal nutrition on hypothalamic neurogenesis was the main topic of several studies. Prenatal HFD/HED exposure in postnatal offspring has been shown to increase neurogenesis and migration of peptide-producing neurons to the hypothalamus [62,80], hypothalamic neurogenesis inhibition [99] and elevated apoptosis of hypothalamic neurons caused by prolonged inflammatory responses [27,100]. Nevertheless, there are numerous studies documenting the impact of perinatal nutrition on morphology and proliferation in the hippocampus and hypothalamus, our study is the first to deal with the connection between maternal exposure to obesogenic diets and alterations of SVZ neurogenesis.

According to our results, the density of proliferating cells within the SVZ and RMS is markedly reduced in mice with elevated body weight and fat (EXF mice) as well as in mice with decreased body weight and body fat (CL mice) in comparison with control mice. Significant enhancement in the number of proliferating cells residing in the SVZ has been reported due to various “brain-healthy diets” based on caloric restriction [101,102] or diets enriched with polyphenols and polyunsaturated fatty acids [103]. The decline in proliferation may be caused by enhanced apoptosis of progenitor cells, but also by deceleration of the cell cycle and induction of cell-cycle arrest.

Interestingly, the decrease in cell proliferation was not equal in examined anatomical sections of the SVZ/RMS region in either CL or EXF mice. The most striking decline was observed in the caudal parts, i.e., in the SVZ and in the vertical arm of the RMS. The differences in responses of the SVZ/RMS anatomical parts to HED can be the result of the higher susceptibility of those parts of the SVZ/RMS, which are characterized by the highest ratio of dividing cells under physiological conditions.

In EXF mice, the reduction in the number of proliferating cells can be linked to a higher incidence of NO producing cells in these mice. NO is an important mediator of antiproliferative effect in neurogenic areas of mammalian brain. Many studies have shown that the inhibition of NO synthesis leads to elevated neuronal precursor proliferation within the SVZ [104,105,106,107]. In the context of these studies we can assume that the decrease in proliferating cell number and simultaneous increase in the NADPH-d-positive cells number in the SVZ/RMS neurogenic region of offspring with elevated body fat and weight can be a demonstration of the antiproliferative effect of NO. To date, signaling pathways and the molecular mechanisms involved in the antiproliferative effect of NO have not been described. However, Carreira et al. [108] have found that proliferation within the SVZ of mice of both sexes can be affected due to decreased function of the epidermal growth factor receptor and this decrease is enhanced by NO from inflammatory origins.

Surprisingly, in CL mice, the cell proliferation was significantly lowered, despite the fact that the total number of NADPH-d+ cells was significantly reduced. Since the CL mice were not exposed to the maternal overfeeding, lower levels of inflammatory signals could prevent the rise of nitrergic cell number. But at the same time, the proliferation was significantly reduced in these mice. In this case, the enhancement of the antiproliferative effect of NO can be related to the better bioavailability of NO. For example, Ungvari et al. [109] have found that caloric restriction increases bioavailability of NO, so it can be hypothesized that the decrease in proliferation activity in CL mice is a result of better utilization of NO, nevertheless the number of NADPH-d+ cells is reduced. 

According to our results, perinatal HED caused a significant increase in the number of degenerating cells within the SVZ/RMS neurogenic area of EXF mice, nevertheless, this increase was more prominent in the CL mice. In opposition to our findings, Niculescu and Lupu [56] reported a slight, non-significant decrease in apoptosis within the area comprising the ventricular zone (VZ) and SVZ in 17 day old embryos exposed to HFD, prior to and during gestation. The discrepancy of our outcomes can be due to the differences in diet composition (HFD vs. HED), to the difference in offspring’s age (embryos vs. adult mice) or to a slight difference in explored area (VZ/SVZ vs. SVZ/RMS). An increase in apoptosis of mature and newly generated neurons after HFD have been previously described also in the hypothalamus of male rats and mice [100,110,111]. On the contrary, apoptosis in the hippocampus of animals perinatally exposed to HFD was decreased [56]. In adult mammalian brain, apoptosis is a supporting element allowing the selection of suitable cells before they complete their differentiation in postnatal life [112], so the changes in cell death rate can be a sign of an eminent neurogenesis malfunction.

Another important finding presented in this study is that changes in proliferation/degeneration of cells were similar in obese and lean mice. The latter were not exposed to HED during perinatal development and this group showed a decline in the amount of body fat and weight, nevertheless, these mice were bred under standardized nutritional conditions and they showed no symptoms of any illness.

Interestingly, in the obesity-induction resistant experimental group of mice characterized by normal weight and moderately elevated body fat deposits (EXN mice), HED during pregnancy and lactation did not result in changes in proliferation, cell degeneration or nitrergic cell number. Specifically, we found that HED decreases cell proliferation and increases the number of degenerating and NO-producing neurons independent of the amount of body fat and body weight of the offspring. Thus, the major finding of this study is that maternal HED affects individual processes of SVZ/RMS neurogenesis only in mice where extreme phenotype, such as significant overweight/adiposity or obesity is manifested. Further, similar changes in neurogenesis are present in normally fed mice, where the lean phenotype spontaneously develops. In contrary to our findings, Val-Laillet et al. [113] showed a reduction in neurogenesis in piglets of both sexes from mothers fed with a Western diet during pregnancy and lactation, although the diet model used in their study was not efficient enough for development of obesity or evident elevation of the body weight and adiposity of adult offspring.

HFD/HED induced obesity models most closely mimic the development of obesity in humans [114]. The diet-induced obesity model can be considered as an interface between nature and nurture. Eating habits and genetic predispositions interact to uncover the hereditary obese phenotype, which appears only in animals exposed to a high fat or high energy diet [115]. In other words, not every animal exposed to HFD/HED will develop an obese phenotype, since genetic factors are crucial in determining an individual response to the environment [116]. Development of the obese phenotype in mice of both sexes is accompanied by changes in the characteristics of the metabolic syndrome of obese humans, such as hyperglycemia, insulin resistance, hypercholesterolemia and hyperleptinemia [117]. Moreover, diet-induced obesity has also been linked to increased inflammation, including microglia activity, in the neurogenic zones, i.e., in the hippocampus [25,118,119] and in the hypothalamus [80]. Microglial-mediated neuroinflammation (i.e., by means of release of inflammatory cytokines) in these neurogenic niches results in neuronal damage and inhibition of proliferation and differentiation of hippocampal/hypothalamic neural stem cells [21,110,120]. On the contrary, according to Park et al. [79], HFD in male mice impairs the proliferation of neural progenitor cells in the dentate gyrus without affecting glial activation and neuronal differentiation, so the reduction in hippocampal neurogenesis is independent of adipose tissue accumulation.

Based on those results, we can hypothesize that the most plausible mechanism responsible for changes in neurogenesis within the SVZ/RMS neurogenic region appears to be enhanced neuroinflammation. The impact of food composition on inflammation within the rodent SVZ was described by Apple et al. [102] in a study on caloric restriction in young and aging animals. In this study, caloric restriction diminished the age-dependent activation of microglia and subsequently decelerated the elevation in pro-inflammatory cytokines resulting in a temporary, enhanced proliferation of neural progenitor cells in young male and female mice.

Besides neuroinflammation, other mechanisms which could be involved in changes of neurogenesis after maternal overnutrition need to be considered. As an outcome of the present study, it could be beneficial to explore the role of growth factors (*brain derived neurotrophic factor*, vascular endothelial growth factor or insulin-like growth factor), especially for the proposed dual role of *brain derived neurotrophic factor (BDNF)*. For instance, in male mice fed with a high-fat diet an impairment of hippocampal neurogenesis alongside a decrease in BDNF levels and elevated lipid peroxidation has been documented [79], contrariwise administration of BDNF to the lateral ventricle did not induce an increase in SVZ neurogenesis in male and female mice, and even decreases neurogenesis in rats of both sexes [121]. In addition to neuroinflammation and growth factors, effects of maternal overfeeding on SVZ/RMS neurogenesis in offspring could be modulated by gut peptides or hormones. For example, the gut peptide ghrelin has a positive effect on all processes of neurogenesis in the SVZ of male and female mice [115,116,122] and the anterior pituitary hormone prolactin enhances SVZ neurogenesis, which leads to an acceleration of the olfactory potential of the mother in preparation for maternal behavior [123].

### 3.3. Mice with Various Amounts of Body Fat Show Comparable Performances in the Hot Plate Test

Given that early life experiences can impact the maturation of the nociceptive system [124] we examined the effects of maternal HED on pain outcomes in adult offspring. According to the literature, poor diet during pregnancy and lactation alters both thermal and mechanical nociceptive responsivity in the offspring [125,126,127]. Besides that, several authors have reported that pain sensitivity changes significantly with increasing body weight [128,129,130,131]. Nevertheless, we didn’t record any significant differences in nociceptive responses to thermal stimuli in mice with various body conditions. Our results can be related to the findings of Tashani et al. [129], which showed that obese individuals were more susceptible to mechanical pain than to thermal pain. Our results are also consistent with the findings of Torensma et al. [128], who reported that overweight subjects showed reduced pain perception to harmful thermal stimuli.

### 3.4. Overfeeding during Prenatal and Early Postnatal Development Enhances Anxiety-Related Behavior, Reduces Locomotor Activity and Impairs Spatial Memory and Learning in Adult Offspring

Beside the substantial effect on offspring, neurogenesis and maternal diet can also affect behavior of offspring in adulthood [132]. Indeed, impairment of spatial learning [133] and elevated anxiety [63,134] have been described, suggesting that maternal HFD/HED may affect brain development, like other stressful events.

Present data revealed that overfeeding of experimental mice during prenatal and early postnatal development mitigated the locomotor activity in offspring with the obese phenotype, but not in offspring with physiological body weight. On the contrary, maternal HED enhanced anxiety-related behavior in offspring regardless of body conditions. The reduction in locomotion activity and the rise in anxiety-like behavior were even more striking in mice with reduced weight and reduced body fat deposits (CL). Similar findings were documented in a study by Ogrodnik et al. [135]. These authors have shown that obese mice of both sexes show increased anxiety-like behavior not related to body mass. According to this study, major contributors to obesity-induced anxiety are the senescent cells, i.e., cells in a non-dividing, altered state into which many vertebrate cells enter when stressed. In humans, maternal overnutrition has been connected with obesity of infants, elevated inflammation and higher rate of anxiety disorders [136,137,138]. In addition to neural inflammation, a perinatal HFD/HED can evoke interruption of anxiety responses through γ-aminobutyric acid, glucocorticoid receptor or by activation of serotonergic and neurotrophin signaling pathways [63,134,139,140].

Besides changes in locomotor activity and anxiety level, adult offspring exposed to the maternal HED showed aggravated spatial learning and memory capability, but this effect was significant only in animals, where the obese phenotype was developed. Learning and memory deficits were evident in the lean mice as well. According to several rodent studies, obesity in pregnant mothers or even consumption of HFD/HED through the perinatal period may cause disturbances in learning abilities and modify performances on the Morris Water Maze [140], Barnes Maze [133], operant conditioning [141] and novel object recognition [142]. The above-mentioned learning disturbances were accompanied by a decline in hippocampal neuronal proliferation, attenuation of BDNF gene expression [143], up regulation of hippocampal lipid peroxidation and amplification of proinflammatory cytokine expression in the young adult and adult hippocampus [23,133,140]. The findings described above indicate that neuroinflammation and oxidative stress may be mediators of cognitive impairment. If these processes are present in the SVZ/RMS neurogenic region of offspring after perinatal overnutrition needs to be examined in future studies.

## 4. Material and Methods

### 4.1. Animal Model

All experiments were performed on female adult mice (Mus musculus) of the outbred ICR strain (Velaz, Prague, Czech Republic). The experiments were carried out in compliance with protocols for animal care, which were accredited by the European Communities Council Directive (2010/63/EU) and with permission of the State Veterinary and Food Administration of the Slovak Republic (4451/14–221, 4247/15–221) under the supervision of the Ethical Council of the Institute of Neurobiology BMC SAS. All experimental procedures were designed and performed in order to reduce the number of animals used and to minimize animal suffering.

An intergenerational dietary model employing overfeeding of experimental mice during prenatal and early postnatal development was applied to acquire mice with various body conditions [144] (Figure 8). Female mice of the parental generation were mated with males of the same strain. Then, the fertilized mice were randomly allocated into two groups:

control group (C, n = 7)—during the gestation period (21 days) and the lactation period (21 days from birth of pups to weaning) the dams were fed a standard pellet diet (M1, Ricmanice, Czech Republic, 3.2 kcal/g) ad libitum.experimental group (EX, n = 12)—during the gestation period (21 days) and the lactation period (21 days from birth of pups to weaning) the dams were fed a standard pellet diet (M1) plus Ensure Plus high-energy nutritional product (Abbott Laboratories, Lake Bluff, IL, USA, 1.5 kcal/mL) ad libitum.

The result of this diet was the origin of offspring (filial generation, F1) with an elevated predisposition for the development of obesity in adulthood. After weaning, all female F1 mice were fed the standard pellet diet only (M1). All animals were housed in a temperature-controlled colony room at 22 ± 1 °C on a reverse 12 h light/dark cycle (6 am to 6 pm) with free access to food and water. On day 34 of their age, F1 mice were individually scanned using MRI (Echo MRI, Whole Body Composition Analyser, Echo Medical System, Houston, TX, USA) in order to evaluate the exact amount of body fat deposits. According to the MRI results, the mice were divided into four groups (CN, CL, EXN, EXF):females acquired from control dams:
➢CN—control group of mice characterized by normal weight (±20g) and physiological percentage of body fat deposits (7–8%).➢CL—control group of mice characterized by reduced weight and reduced body fat deposits (<7%).females acquired from experimental dams:
➢EXN—experimental group of mice characterized by normal weight and moderately elevated body fat deposits (8–11%), obesity-induction resistant experimental mice.➢EXF—experimental group of mice characterized by markedly elevated weight (±24 g) and body fat deposits (>11%).

In the time of sexual adulthood (starting Day 42 of age), animals from all four groups were processed for behavioral testing, histochemical and immunohistochemical evaluation.

For complete feeding parameters of dams and their offspring, complete somatic parameters of dams and their offspring and for analysis of blood parameters of F1 generation see Kubandová et al., 2014 [144].

### 4.2. Open Field

In order to document the locomotor activity and the anxiety-related behavior in mice with various amounts of body fat, we employed the open field test as described previously [145,146]. The OF was a black square box without a ceiling (60 cm × 45 cm × 35 cm). A computer connected video camera was hanged above the OF (1.20 m above the surface of the testing apparatus). At the beginning of the trial, the mouse was laid on the middle point of the OF and tested for 8 min. During testing, the researcher left the testing room. Ethovision XT 7 was used to analyze the distance traveled and the mean velocity to measure the locomotor activity of mice. To measure the anxiety level, all behaviors (grooming, rearing, freezing, the number of crossing the OF middle point, the time spent by the OF walls and defecation, thigmotaxis) were scored from video files by an experimenter who was blind to the experimental conditions. To explore neophobia, the latency to leave the peripheral compartment for the first time was recorded.

### 4.3. Hot Plate Test

To measure thermal nociception in mice a standard hot plate test was used [147]. The mice were placed on a metal plate heated to 54 °C and a transparent plastic cylinder (diameter 18 cm, height 27 cm) was used to retain the animals on the apparatus. In this test, hind paw response (licking or shaking) or jumping was regarded as the pain threshold. The latency to lick a paw or jump off (in seconds) was recorded. Right after the response, the mice were quickly removed from the apparatus. Termination time for this test was 60 s.

### 4.4. Barnes Maze

To measure spatial navigation ability in mice with various amounts of body fat the Barnes maze was used [148]. The Barnes maze was a round elevated platform (90 cm diameter, 100 cm height) with 20 usable escape holes (5 cm diameter) situated at the periphery. A black acrylic escape box (20 cm × 10 cm × 8 cm; L × W × H) was hidden under single hole. The apparatus was placed under bright overhead lighting in a room with four visual cues at the periphery. Over 5 days, all of the mice underwent 20 acquisition trials. On day six the 3 min investigatory trial was carried out. Acquired video files were scored by a researcher, who was blind to the experimental conditions. The latency to find the target hole and number of reference errors (entering the incorrect hole) were documented.

### 4.5. Tissue Processing

Mice that were 49 days old from all four groups (12 animals per group, 48 animals total) were deeply anesthetized (isoflurane and chloralhydrate) and perfused transcardially with 4% paraformaldehyde (PFA) in 0.1 M PBS (phosphate-buffered saline). The brains were extracted from the skulls. After overnight post-fixation in 4% PFA in 0.1 M PB (phosphate buffer), the brains were placed into 30% sucrose in 0.1 M PBS for cryoprotection. Afterwards, sagittal sections (30 μm) were cut on a cryomicrotome and embedded in dishes with 0.1 M PBS. Thereafter, the sections were processed for histochemical and immunohistochemical analyses.

To assess the effects of maternal diet on cell proliferation in the SVZ/RMS the sections were stained with the proliferation marker Ki-67 as described previously [74]. To visualize the proliferating cells, the anti-Ki-67 antibody (1:1000, Dako, for 18 h) and biotinylated goat anti-rabbit IgG (1:200, Santa Cruz, CA, USA) secondary antibody for 2 h at room temperature were used.

NADPH-diaphorase (NADPH-d) reaction was performed as described previously [149]. 0.1 M PB, pH 7.4 containing 0.4 mg/mL nitroblue tetrazolium, 0.3% Triton X-100, 5 mg/mL malic acid, 4 mg/mL magnesium chloride and 0.8 mg/mL NADPH-d at 37 °C for 1 h was used for visualization of differentiated cells.

For detection of degenerating cells, fluoro-jade C (FJC) was used according to standard protocols [149]. 0.01% FJC stock solution (10 mg of dye powder + 100 mL of staining solution) was used to prepare the staining medium (1 mL of stock solution + 99 mL of 0.1% CH_3_COOH vehicle.

The shape and the thickness of the SVZ/RMS were evaluated on hematoxylin-eosin stained sections. The staining was conducted according to standard protocols [150].

### 4.6. Image Analysis and Quantification

To explore the morphological characteristics of the SVZ/RMS, as well as the NADPH-d and Ki-67 positivity in the mouse SVZ/RMS, a light microscope (Olympus BX51) equipped with a DP50camera system was used. To explore the prevalence of FJC+ cells, the Olympus Reflected Fluorescence System U-RFL-T was employed. Only sections in which the entire range of RMS was visible were evaluated (six sections for each animal). To evaluate the hematoxylin-eosin sections, images were captured. The thickness of the SVZ and RMS was measured with UTHSCSA Image Tool (version 3.0). The NADPH-d + cells were counted along the whole SVZ and individually in three anatomical parts of the RMS—the vertical arm, elbow and horizontal arm. To evaluate the number of Ki-67+ cells and FJC+ cells in the SVZ/RMS, sections were viewed under a × 20 objective, and a visual grid was used to manually count the number of Ki-67+/FJC+ neurons in each region (200 µm × 200 µm area). If the researched area was larger than the borders of the grid, in order to count proliferating/degenerating cells, the grid was fixed in the investigated region according to familiar anatomical benchmarks. If the area to be analyzed was smaller than the borders of the grid, only the area of interest was counted. GraphPad Prism 5.0 (GraphPad Software Inc., San Diego, CA, USA) was used for statistical analysis of all behavioral and immunohistochemical data. All data were analyzed with the one-way ANOVA test and Tukey–Kramer test. The values were expressed as means ± standard error of the mean (SEM). Significant changes are labeled as * *p* ≤ 0.05, ** *p* ≤ 0.01, *** *p* ≤ 0.001.

## 5. Summary

Overall, our findings demonstrate an intergenerational effect of maternal high-energy nutrition on individual processes of adult neurogenesis in the SVZ/RMS olfactory neurogenic region and some behavioral tasks including memory, anxiety and nociception. The major finding of this study is that HED affects neurogenesis and cognition only in mice where an extreme phenotype, such as significant overweight/adiposity or obesity is manifested. Interestingly, maternal HED enhanced anxiety-related behavior in offspring regardless of its body conditions. Surprisingly, changes in neurogenesis and learning and memory deficits were also evident in mice spontaneously displaying a lean phenotype, which were not exposed to HED during perinatal development. Further investigation of mechanisms by which prenatal and early-life nutritional factors multigenerationally influence SVZ/RMS neurogenesis, anxiety and cognitive function would contribute to understanding the hazardous effect of maternal HED nutrition on SVZ/RMS neurogenesis and some behavioral tasks.

## Figures and Tables

**Figure 1 ijms-23-05564-f001:**
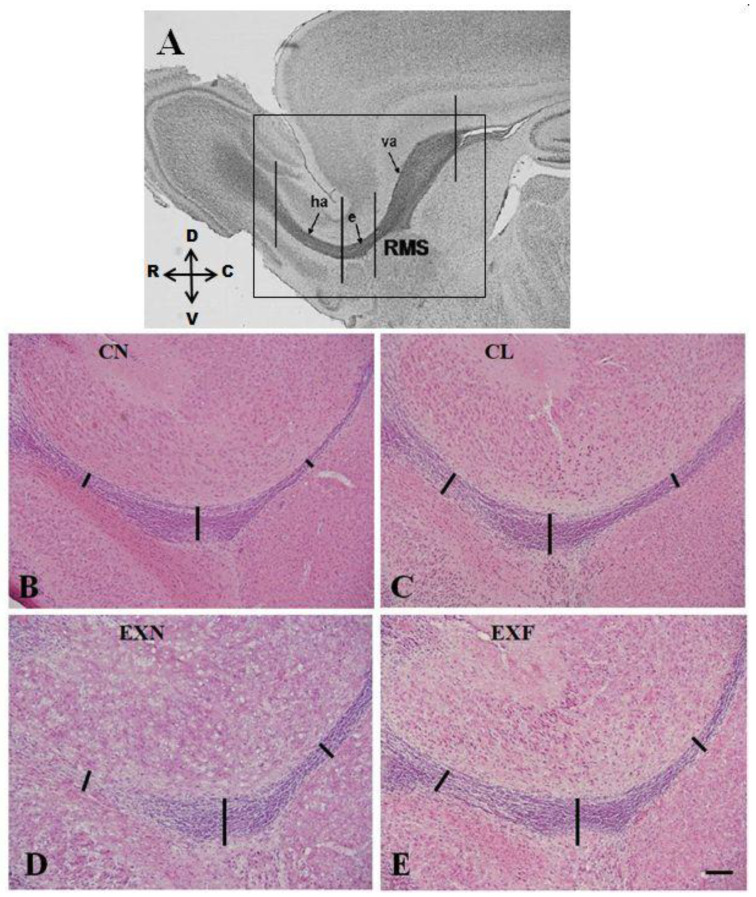
Microphotographs of parasagittal sections of the RMS of mice with various body conditions stained with hematoxylin-eosin. (**A**) Micrograph showing individual anatomical parts of the RMS: va—vertical arm, e—elbow, ha—horizontal arm. (**B**–**E**) The dark violet staining, an indicator of cell density distinguishes the RMS from other areas of the forebrain. Note the increase in the thickness of the RMS caudal parts (the vertical arm and the elbow) in CL, EXN and EXF mice when compared to controls (black lines). Abbreviations: (CN) control group of mice characterized by normal weight and physiological percentage of body fat deposits; (CL) control group of mice characterized by reduced weight and reduced body fat deposits; (EXN) experimental group of mice characterized by normal weight and moderately elevated body fat deposits; (EXF) experimental group of mice characterized by markedly elevated weight and body fat deposits; RMS—rostral migratory stream, C-caudal, R-rostral, D-dorsal, V-ventral. Scale bar = 100 μm.

**Figure 2 ijms-23-05564-f002:**
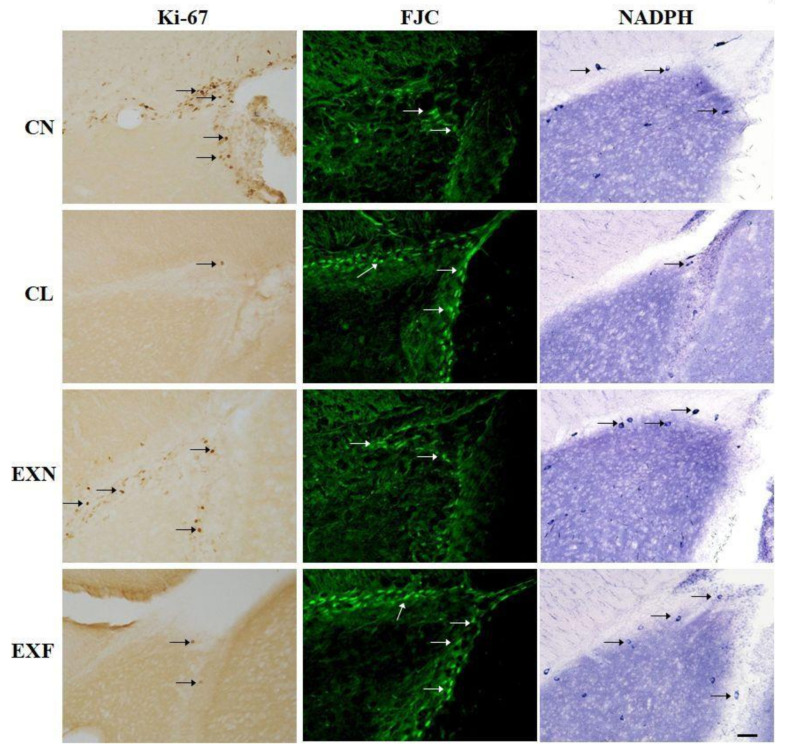
Changes in adult neurogenesis in the subventricular zone of mice with various body conditions. (column one) Immunohistochemical staining of proliferating cells by means of Ki-67 (*brown*) was performed to examine the effect of HED on cell proliferation in the SVZ/RMS; (column two) fluoro-jade C labelling was performed to identify degenerating cells (*green*); (column three) effect of maternal overnutrition on the distribution of nitrergic cells, arrows stand for the NADPH-d-positive cells. Abbreviations: (CN) control group of mice characterized by normal weight and physiological percentage of body fat deposits; (CL) control group of mice characterized by reduced weight and reduced body fat deposits; (EXN) experimental group of mice characterized by normal weight and moderately elevated body fat deposits; (EXF) experimental group of mice characterized by markedly elevated weight and body fat deposits. Scale bar = 50 μm.

**Figure 3 ijms-23-05564-f003:**
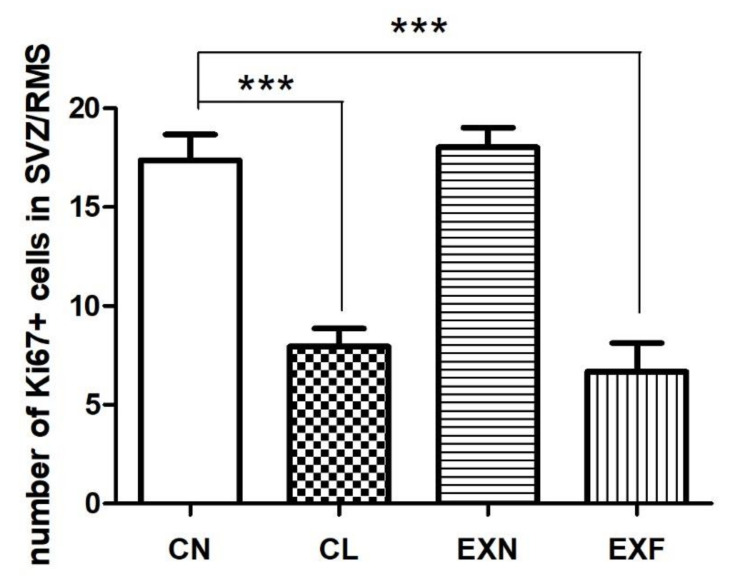
The total number of Ki-67+ cells in the SZV/RMS neurogenic region in mice with various body conditions. Abbreviations: SVZ—subventricular zone; RMS—rostral migratory stream; (CN) control group of mice characterized by normal weight and physiological percentage of body fat deposits; (CL) control group of mice characterized by reduced weight and reduced body fat deposits; (EXN) experimental group of mice characterized by normal weight and moderately elevated body fat deposits; (EXF) experimental group of mice characterized by markedly elevated weight and body fat deposits. Data are shown as mean ± SEM. Statistical significance of differences between experimental and control groups: *** *p* ≤ 0.001.

**Figure 4 ijms-23-05564-f004:**
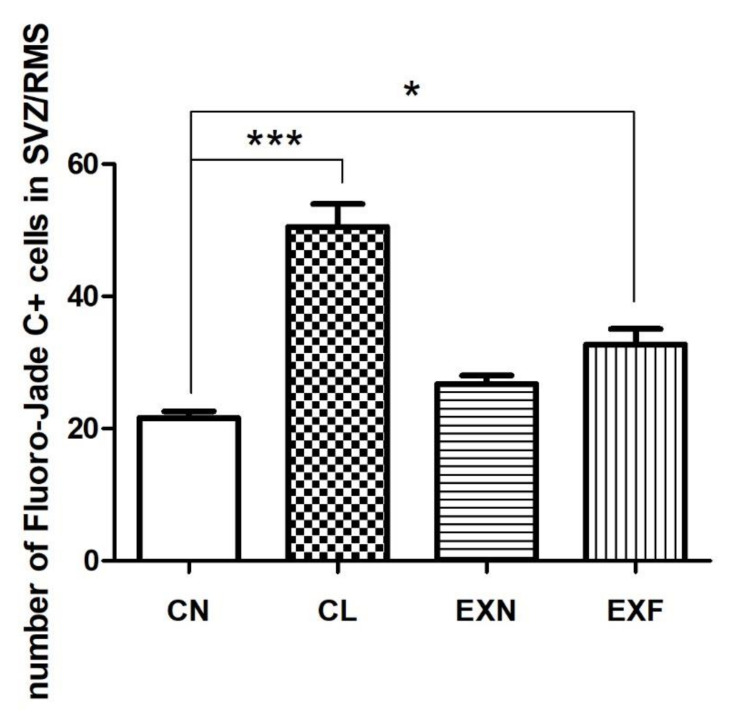
The total number of FJC+ cells in the SZV/RMS neurogenic region in mice with various body conditions. Abbreviations: SVZ—subventricular zone; RMS—rostral migratory stream; (CN) control group of mice characterized by normal weight and physiological percentage of body fat deposits; (CL) control group of mice characterized by reduced weight and reduced body fat deposits; (EXN) experimental group of mice characterized by normal weight and moderately elevated body fat deposits; (EXF) experimental group of mice characterized by markedly elevated weight and body fat deposits. Data are shown as mean ± SEM. Statistical significance of differences between experimental and control groups: * *p* ≤ 0.05, *** *p* ≤ 0.001.

**Figure 5 ijms-23-05564-f005:**
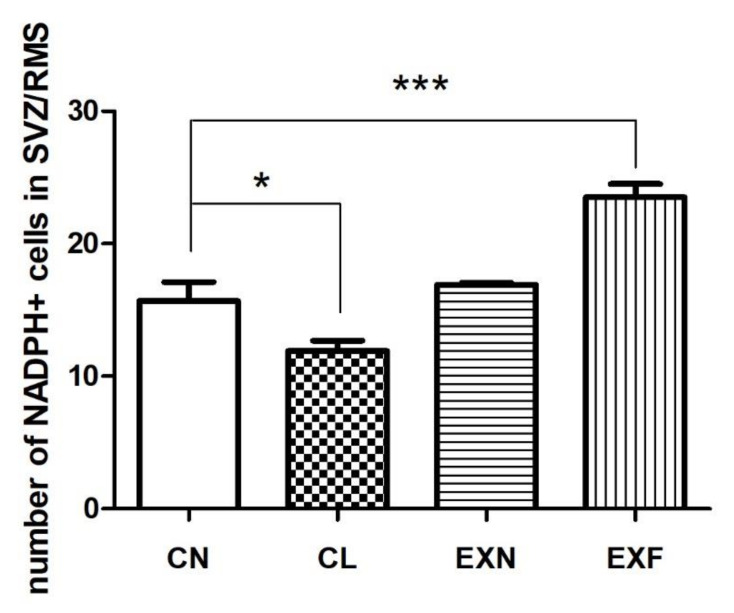
The total number of NADPH-d+ cells in the SZV/RMS neurogenic region in mice with various body conditions. Abbreviations: SVZ—subventricular zone; RMS—rostral migratory stream; (CN) control group of mice characterized by normal weight and physiological percentage of body fat deposits; (CL) control group of mice characterized by reduced weight and reduced body fat deposits; (EXN) experimental group of mice characterized by normal weight and moderately elevated body fat deposits; (EXF) experimental group of mice characterized by markedly elevated weight and body fat deposits. Data are shown as mean ± SEM. Statistical significance of differences between experimental and control groups: * *p* ≤ 0.05, *** *p* ≤ 0.001.

**Figure 6 ijms-23-05564-f006:**
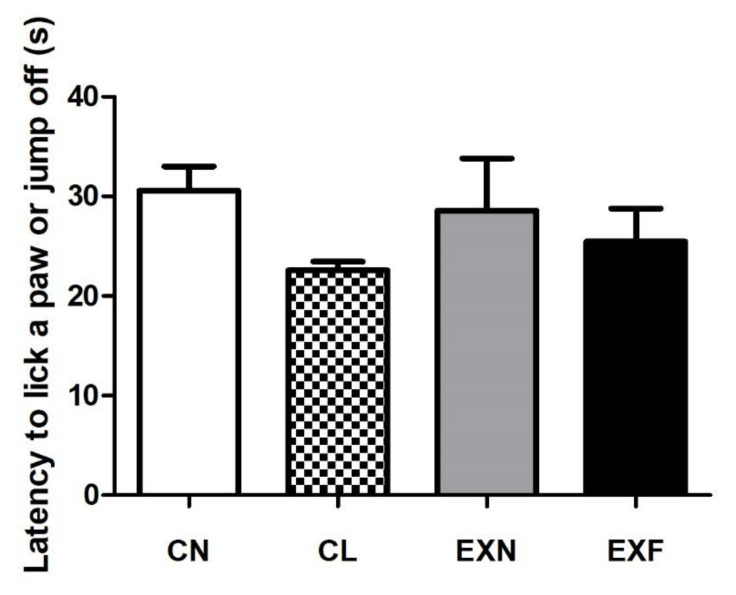
Latency to lick a paw or jump off (s) in mice with various body conditions. Abbreviations: (CN) control group of mice characterized by normal weight and physiological percentage of body fat deposits; (CL) control group of mice characterized by reduced weight and reduced body fat deposits; (EXN) experimental group of mice characterized by normal weight and moderately elevated body fat deposits; (EXF) experimental group of mice characterized by markedly elevated weight and body fat deposits. Data are shown as mean ± SEM.

**Figure 7 ijms-23-05564-f007:**
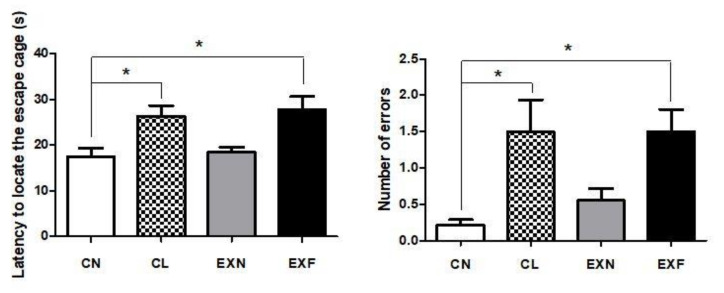
The latency to find the target hole (s, left) and number of reference errors (entering the incorrect hole, right) in mice with various body conditions. Abbreviations: (CN) control group of mice characterized by normal weight and physiological percentage of body fat deposits; (CL) control group of mice characterized by reduced weight and reduced body fat deposits; (EXN) experimental group of mice characterized by normal weight and moderately elevated body fat deposits; (EXF) experimental group of mice characterized by markedly elevated weight and body fat deposits. Statistical significance of differences between experimental and control groups: * *p* ≤ 0.05. Data are shown as mean ± SEM.

**Figure 8 ijms-23-05564-f008:**
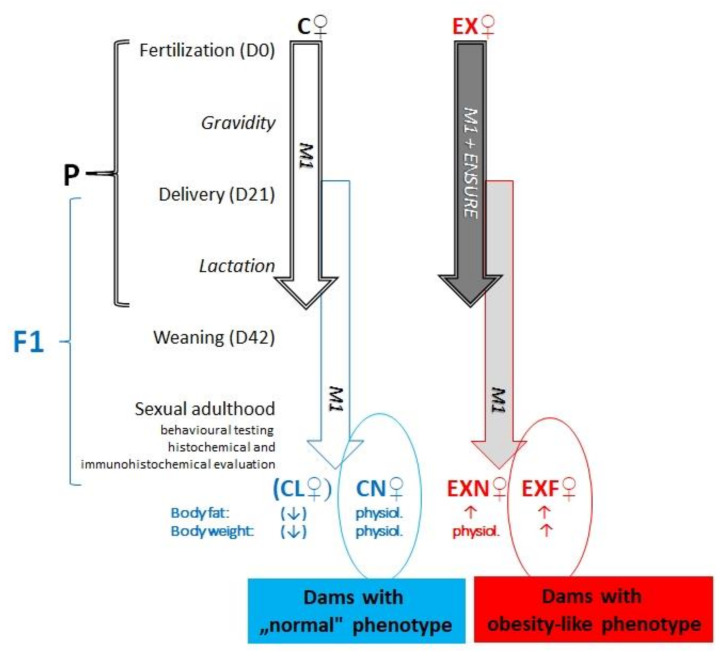
Intergenerational dietary model. Scheme depicts the dietary regime of two generations of mice used in this study. Abbreviations: P—parental generation; F1—the first filial generation; M1—standard pellet diet, ENSURE—diet supplement with Ensure PLUS; C—control group of mice; EX—experimental group of mice; (CN) control group of mice characterized by normal weight and physiological percentage of body fat deposits; (CL) control group of mice characterized by reduced weight and reduced body fat deposits; (EXN) experimental group of mice characterized by normal weight and moderately elevated body fat deposits; (EXF) experimental group of mice characterized by markedly elevated weight and body fat deposits.

**Table 1 ijms-23-05564-t001:** Thickness of the SVZ/RMS in mice with various body conditions.

Group (% of Body Fat)	CN (Control; 7–8%) (n = 6)	CL (<7%) (n = 6)	EXN (8–11%) (n = 6)	EXF (>11%) (n = 6)
Brain Area				
SVZ (µm)	6.80 ± 1.03	7.37 ± 0.54	8.23 ± 0.74	13.70 ± 1.14 ***
vertical arm of the RMS (µm)	3.47 ± 0.35	5.40 ± 0.62 *	5.27 ± 0.31 *	5.32 ± 0.45 *
elbow of the RMS (µm)	18.13 ± 0.51	20.83 ± 0.73 *	20.27 ± 0.23 *	20.48 ± 0.59 *
horizontal arm of the RMS (µm)	9.95 ± 0.85	8.43 ± 0.25	8.10 ± 1.10	9.38 ± 0.44

Abbreviations: (CN) control group of mice characterized by standard weight and physiological percentage of body fat deposits; (CL) control group of mice characterized by reduced weight and reduced body fat deposits; (EXN) experimental group of mice characterized by standard weight and moderately elevated body fat deposits; (EXF) experimental group of mice characterized by markedly elevated weight and body fat deposits; SVZ—subventricular zone; RMS—rostral migratory stream. Data are shown as mean ± SEM. Statistical significance of differences between experimental and control groups: * *p* ≤ 0.05, *** *p* ≤ 0.001.

**Table 2 ijms-23-05564-t002:** The number of Ki-67+ cells in the SVZ/RMS neurogenic region (200 µm × 200 µm area) in mice with various body conditions. Abbreviations: SVZ—subventricular zone; RMS—rostral migratory stream; (CN) control group of mice characterized by normal weight and physiological percentage of body fat deposits; (CL) control group of mice characterized by reduced weight and reduced body fat deposits; (EXN) experimental group of mice characterized by normal weight and moderately elevated body fat deposits; (EXF) experimental group of mice characterized by markedly elevated weight and body fat deposits. Data are shown as mean ± SEM. Statistical significance of differences between experimental and control groups: * *p* ≤ 0.05, *** *p* ≤ 0.001.

Group (% of Body Fat)	CN (Control; 7–8%) (n = 6)	CL (<7%) (n = 6)	EXN (8–11%) (n = 6)	EXF (>11%) (n = 6)
Brain Area				
SVZ	6.66 ± 0.62	1.90 ± 0.44 ***	5.18 ± 0.43	1.98 ± 0.62 ***
vertical arm of the RMS	3.04 ± 0.55	1.60 ± 0.45	2.62 ± 0.20	1.13 ± 0.47 *
elbow of the RMS	5.50 ± 1.01	2.61 ± 0.24 *	5.36 ± 0.70	2.23 ± 0.52 *
horizontal arm of the RMS	2.88 ± 0.40	1.89 ± 0.15	2.13 ± 0.55	1.09 ± 0.26 *

**Table 3 ijms-23-05564-t003:** The number of FJC+ cells in the SVZ/RMS neurogenic region (200 µm × 200 µm area) in mice with various body conditions. Abbreviations: SVZ—subventricular zone; RMS—rostral migratory stream; (CN) control group of mice characterized by normal weight and physiological percentage of body fat deposits; (CL) control group of mice characterized by reduced weight and reduced body fat deposits; (EXN) experimental group of mice characterized by normal weight and moderately elevated body fat deposits; (EXF) experimental group of mice characterized by markedly elevated weight and body fat deposits. Data are shown as mean ± SEM. Statistical significance of differences between experimental and control groups: *** *p* ≤ 0.001.

Group (% of Body Fat)	CN (Control; 7–8%) (n = 6)	CL (<7%) (n = 6)	EXN (8–11%) (n = 6)	EXF (>11%) (n = 6)
Brain Area				
SVZ	6.72 ± 1.16	17.79 ± 1.49 ***	7.98 ± 0.57	19.87 ± 1.50 ***
vertical arm of the RMS	5.13 ± 0.57	10.55 ± 0.97 ***	5.30 ± 0.40	2.77 ± 0.39
Elbow of the RMS	5.85 ± 0.37	14.25 ± 1.43 ***	8.23 ± 0.79	5.43 ± 0.67
Horizontal arm of the RMS	3.87 ± 0.36	8.53 ± 0.84 ***	5.55 ± 0.56	3.82 ± 0.82

**Table 4 ijms-23-05564-t004:** The number of NADPH-d+ cells in the SVZ/RMS neurogenic region in mice with various body conditions. Abbreviations: SVZ—subventricular zone; RMS—rostral migratory stream; (CN) control group of mice characterized by normal weight and physiological percentage of body fat deposits; (CL) control group of mice characterized by reduced weight and reduced body fat deposits; (EXN) experimental group of mice characterized by normal weight and moderately elevated body fat deposits; (EXF) experimental group of mice characterized by markedly elevated weight and body fat deposits. Data are shown as mean ± SEM. Statistical significance of differences between experimental and control groups: * *p* ≤ 0.05, ** *p* ≤ 0.01.

Group (% of Body Fat)	CN (Control; 7–8%) (n = 6)	CL (<7%) (n = 6)	EXN (8–11%) (n = 6)	EXF (>11%) (n = 6)
Brain Area				
SVZ	9.74 ± 1.27	8.62 ± 0.68	12.55 ± 0.42	14.06 ± 1.22 *
vertical arm of the RMS	1.64 ± 0.06	1.62 ± 0.10	1.55 ± 0.42	1.97 ± 0.16
elbow of the RMS	1.63 ± 0.35	1.51 ± 0.24	2.43 ± 0.08	2.78 ± 0.19 *
horizontal arm of the RMS	2.25 ± 0.23	1.58 ± 0.34	1.65 ± 0.07	3.70 ± 0.23 **

**Table 5 ijms-23-05564-t005:** Behavioral scores for locomotor activity in open field test in mice with various body conditions. Data are shown as mean ± SEM. Abbreviations: (CN) control group of mice characterized by normal weight and physiological percentage of body fat deposits; (CL) control group of mice characterized by reduced weight and reduced body fat deposits; (EXN) experimental group of mice characterized by normal weight and moderately elevated body fat deposits; (EXF) experimental group of mice characterized by markedly elevated weight and body fat deposits. Statistical significance of differences between experimental and control groups: * *p* ≤ 0.05.

Group (% of Body Fat)	CN (Control; 7–8%) (n = 6)	CL (<7%) (n = 6)	EXN (8–11%) (n = 6)	EXF (>11%) (n = 6)
Behavioral Measures				
The distance travelled (cm)	3666 ± 198.10	2737 ± 143.70 *	3179 ± 234.10	2846 ± 212.40 *
Mean velocity (cm/s)	11.74 ± 0.64	10.55 ± 1.00	11.09 ± 0.97	10.62 ± 0.79

**Table 6 ijms-23-05564-t006:** Behavioral scores for locomotor activity in the Barnes maze in mice with various body conditions. Data are shown as mean ± SEM. Abbreviations: (CN) control group of mice characterized by normal weight and physiological percentage of body fat deposits; (CL) control group of mice characterized by reduced weight and reduced body fat deposits; (EXN) experimental group of mice characterized by normal weight and moderately elevated body fat deposits; (EXF) experimental group of mice characterized by markedly elevated weight and body fat deposits. Statistical significance of differences between experimental and control groups: * *p* ≤ 0.05.

Group (% of Body Fat)	CN (Control; 7–8%) (n = 6)	CL (<7%) (n = 6)	EXN (8–11%) (n = 6)	EXF (>11%) (n = 6)
Behavioral Measures				
The distance travelled (cm)	200.70 ± 17.04	132.7 ± 14.68 *	211.30 ± 21.36	140.70 ± 8.13 *
Mean velocity (cm/s)	13.54 ± 084	12.53 ± 1.20	13.39 ± 1.07	12.52 ± 0.99

**Table 7 ijms-23-05564-t007:** Behavioral scores for anxiety-related behavior in mice with various body conditions. Abbreviations: (CN) control group of mice characterized by normal weight and physiological percentage of body fat deposits; (CL) control group of mice characterized by reduced weight and reduced body fat deposits; (EXN) experimental group of mice characterized by normal weight and moderately elevated body fat deposits; (EXF) experimental group of mice characterized by markedly elevated weight and body fat deposits. Data are shown as mean ± SEM. Statistical significance of differences between experimental and control groups: * *p* ≤ 0.05, ** *p* ≤ 0.01, *** *p* ≤ 0.001.

Group (% of Body Fat)	CN (Control; 7–8%) (n = 6)	CL (<7%) (n = 6)	EXN (8–11%) (n = 6)	EXF (>11%) (n = 6)
Anxiety Scores				
Number of grooms	2.00 ± 0.37	1.83 ± 0.31	1.50 ± 0.22	1.17 ± 0.17
Grooming (s)	13.83 ± 0.65	12.50 ± 1.69	9.83 ± 2.47	9.33 ± 1.05
Latency to groom (s)	72.94 ± 2.98	154.4 ±5.27 ***	71.57 ± 5.66	145.0 ± 6.12 ***
Number of rears	60.50 ± 3.34	31.67 ± 5.53	42.33 ± 4.35 *	41.83 ± 2.41 *
Rearing (s)	72.67 ± 2.45	42.50 ± 5.14 **	53.50 ± 6.85	62.83 ± 4.20
Number of freezes	3.5 ± 0.43	18 ± 4.32 ***	6.7 ± 0.81	15.3 ± 3.27 **
Neophobia (s)	42.32 ± 2.45	139.19 ± 4.19 ***	39.31 ± 1.45	142.23 ± 5.43 ***
Middle point crossings	8.17 ± 1.28	2.17 ± 0.48 ***	4.00 ± 0.77 *	3.83 ± 0.83 *
Time spent by the walls (s)	3.10 ± 0.28	3.79 ± 0.15	5.81 ± 0.19 ***	7.78 ± 0.23 ***
Thigmotaxis	1.57 ± 0.25	1.03 ± 0.08	0.33 ± 0.03 ***	0.05 ± 0.01 ***
Defecation	0.50 ± 0.22	2.83 ± 0.65 *	1.83 ± 0.65	2.50 ± 0.22 *

## Data Availability

All data generated or analyzed during this study are included in this published manuscript. All datasets collected during the present study are available from the corresponding author on reasonable request.

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
