# Peer review of "Maternal High-Energy Diet during Pregnancy and Lactation Impairs Neurogenesis and Alters the Behavior of Adult Offspring in a Phenotype-Dependent Manner"

_ijms, 2022, doi:10.3390/ijms23105564_

Round 1

Reviewer 1 Report

-The authors present a well detailed analysis of the effect of pre/postnatal HCD on SVZ and RMS neurogenesis and proliferation. A few points to consider:

-the use of "inadequate" invokes a meaning of under-feeding while I think you mean inadequate in terms of a HCD.  Perhaps using another word here consistent with your introduction of over-feeding or high calorie intake would be more clear?

-as this is a journal that requires results before methods, perhaps some small methodological details in the results to help understand them would be useful such as stating the groups (CN, CL, etc) in the beginning of the results.

-suggest defining abbreviations as they first arrive (e.g., "EXF")

-there are a number of statistical tests, although many have a very low p-value which makes this limitation less concerning, was there a procedure to control for multiple tests or perhaps at least as a limitation (limitations are not considered in discussion and should be listed as well).

-although thorough, the discussion should be shortened if at all possible.

-given the ad libitum diets, can you present statistics on the food intake in the respective EX and C groups?

Author Response

Dear reviewer,

We would like to thank you for the comments concerning our manuscript. We carefully addressed all the comments and according to the advice changes were included in the revised manuscript.

Reviewer 2 Report

The authors present a complex experimental design to assess the impact of maternal high-energy diet in the adult offsprings at neuromorfological and behavioral level. The results have an important translational value in this field. In the following sentence, some comments and questions are indicated in a constructive manner with the aim to provide some gain of research outputs of the current study and the understanding of the topic

Introduction
Lines 38-40: References of epidemiological data should be provided.

Lines 47-56 and 57-  In introducing the conceptual definition of high-caloric diets (HCD) would be better to follow the one of high-energy diet (HED) since the impact on mental health reporting at the epidemiological level for HCD can overlap (not here, in the references provided, but in the literature) with the one due to HED. For instance, the increased risk or AD or mood disorders is due to HCD or specifically attributable to HED, or both? This should be clarified.

Results in the open-field test  Table 6
The results shoulds of the anxiety-like profile of animals should also report the ‘neophobia’ not just a ‘number of freezes’, the latency of grooming (not only the number and time) and the thigmotaxis (ratio central vs periphery), since the report of these variables is missing.

Please, indicate the characteristics of the barns maze with respect to size of holes and cage that may influence the animal’s performance in the maze.
Please, report the locomotor activity performed in the barns test.

Is there any correlation between the behavioral performances and the other variables studied, mostly with the data of the animals with extreme phenotypes?

Since the authors surprisingly found also changes in neurogenesis and L&M in the lean phenotype, probably the sentence 21-23 is more appropriate as the starting point of the abstract and work.

Nothing is said about the estrous cycle of the females.

Please, discuss the work with respect to sex/gender medicine 

Author Response

Dear reviewer,

We would like to thank you for your comments, which we appreciate very much. In accordance with your comments, a revision of the MS was done and all remarks, raised by you, were considered.

Round 2

Reviewer 2 Report

The authors have taken all the comments and queries under consideration, and have send a rebuttal letter with agreements or justifications of their criteria to disagree, that are acceptable and coherent.

The authors have improved the methodological description as requested, and most importantly, they have done an effort to re-analyse the behaviors, providing new variables and data to the results that increase the behavioral characterization and understanding of the behavioral implications of their study. This has also been added in the discussion.

They have also done an effort to dissect the sex issue in the literature discussed and have indicated when the results apply to males, females or both sexes. This is important to be done and it will help to fill in the gap that exists in this respect. Their work was performed in females, and their statement on the consequent limitations is adequate.

Therefore, the review process has been satisfactory and in my opinion does not need further discussion.

I thank the chance to review the work and provide some questions important to be considered.